# Effects of inbound attendees of a mass gathering event on the COVID-19 epidemic using individual-based simulations

**Masaya M. Saito**[1]*, **Kotoe Katayama**[2], **Akira Naruse**[3], **Peiying Ruan**[4], **Michio Murakami**[5], **Tomoaki Okuda**[6], **Tetsuo Ysutaka**[7], **Wataru Naito**[8], **Masaharu Tsubokura**[9], **Seiya Imoto**[2,10]

**1** Department of Information Security, Faculty of Information Systems, University of Nagasaki, Siebold, Manabino, Nagayocho, Nishisonogigun, Nagasaki, Japan, **2** Laboratory of Sequence Analysis, Human Genome Center, The Institute of Medical Science, The University of Tokyo, Shirokanedai, Minato-ku, Tokyo, Japan, **3** NVIDIA, Santa Clara, California, United States of America, **4** NVIDIA AI Technology Center, Tokyo, Japan, **5** Center for Infectious Disease Education and Research (CiDER), Osaka University, Yamadaoka, Suita, Osaka, Japan, **6** Department of Applied Chemistry, Faculty of Science and Technology, Keio University, Hiyoshi, Kohoku, Yokohama, Kanagawa, Japan, **7** Institute for Geo-Resources and Environment, National Institute of Advanced Industrial Science and Technology (AIST), Higashi, Tsukuba, Ibaraki, Japan, **8** Research Institute of Science for Safety and Sustainability, National Institute of Advanced Industrial Science and Technology (AIST), Onogawa, Tsukuba, Ibaraki, Japan, **9** Department of Radiation Health Management, Fukushima Medical University School of Medicine, Hikarigaoka, Fukushima, Fukushima, Japan, **10** Division of Health Medical Intelligence, Human Genome Center, The Institute of Medical Science, The University of Tokyo, Shirokanedai, Minato-ku, Tokyo, Japan

* saitohm@sun.ac.jp

## Abstract

Given that mass gathering events involve heterogeneous and time-varying contact between residents and visitors, we sought to identify possible measures to prevent the potential acceleration of the outbreak of an emerging infectious disease induced by such events. An individual-based simulator was built based on a description of the reproduction rate among people infected with the infectious disease in a hypothetical city. Three different scenarios were assessed using our simulator, in which controls aimed at reduced contact were assumed to be carried out only in the main event venue or at subsequent additional events, or in which behavior restrictions were carried out among the visitors to the main event. The simulation results indicated that the increase in the number of patients with COVID-19 could possibly be suppressed to a level equivalent to that if the event were not being held so long as the prevalence among visitors was only slightly higher than that among domestic residents and strict requirements were applied to the activities of visitors.

## 1. Introduction

Mass gathering events, such as the Olympic games, world expositions, and music festivals, involve large-scale mixing of people from different regions for extended durations (e.g., durations of a few days to several months). Motivated by the Tokyo 2020 Olympic Games, which were eventually held at the end of July 2021 amidst the coronavirus disease 2019 (COVID-19)

**Data availability statement:** This is a simulation-based work. Program codes are available via https://github.com/mmsaito/dist-ibm-covid19 and parameter setting for them are stated in the manuscript. As for a detailed procedure to actually run simulator and interpret the result, please see how_to_run_simulator.docx in this repository. For clearity, we here declare URL https://github.com/mmsaito/dist-ibm-covid19 as the name of the repository.

**Funding:** One of the authors, Masaya M. Saito was supported by JSPS KAKENHI Grant Numbers JP18K11541. This funder had no role in study design, data collection and analysis, decision to publish, or preparation of the manuscript.

pandemic, we aimed to understand the degree to which the reproduction rate can be accelerated by such heterogeneous mixing or be suppressed by behavioral changes from a dynamical perspective. Although an environmental exposure model can quantitatively provide infection risks for various situations [1,2] and be applied to mass gathering events [3], tracking potential infection events using a mechanistic model could be more effective for devising policies and allowing the general population to understand how such interventions work. For such a mechanistic description, an individual-based (or agent-based) model is known to be useful [4].

The application of agent-based models can be traced back to problems involving the containment of a potential pandemic caused by the novel influenza A virus subtype H5N1 in southeast Asia [5,6]. Ferguson et al. [7] also adjusted their framework to the COVID-19 epidemic in the UK and US in the early stage when non-pharmaceutical interventions were the only available measures and vaccines had yet to be developed. Their simulation results suggested that flattening the epidemic curve was not sufficient to avoid a surge of patients exceeding the capacity of hospitals, and thus, extensive non-pharmaceutical interventions combining home quarantine and social distancing should be administrated while closely monitoring their impacts on the economy and social activities, to reduce the effective reproduction number ($R_t$) below unity. Referring to mobility and census data, Aleta et al. [8] constructed a detailed agent-based model of transmission in the Boston metropolitan area, and found that social distancing followed by testing and contact tracing could keep the number of cases below the limit of medical capacity. An individual-based model is considered more advantageous than a compartmental one (e.g., the susceptible, infected, and removed or recovered SIR-like model) when an infectious agent spreads among different communities, as it is highly stochastic in the early stage of the epidemic and keen to provide a correct description of the transmission dynamics. The most of infected people, in general, lose their infectiousness without yielding the successor generations, which are attributable to quite small portion of the predecessors, the so-called super spreaders. The super spreaders are visible in a small outbreak; in MERS 2015 nosocomial outbreak, for example, these were identified as the starting nodes of transmission chain [9] and a single component having a significantly higher reproduction number than the rest of components in the next generation matrix [10]. There possibly exist common risk factors of contracting SARS-CoV-2 behind the superspreading, and a multinational cross-sectional study [11] found as significant factors, history of being infected to SARS-CoV or MERS-CoV, habit of hugging when greeting, and smoking. Son et al. [12] successfully predicted the number of cases that would arise in Daegu, South Korea by using an agent-simulation model, where the majority of reported cases at that time (end of March 2020) were tied to a single church and the transition from the church to other communities was a key factor in the prediction. Religion-related mass gatherings were held also in other countries during a pandemic wave growing; the Kumbh Mela celebration lasted for one month since April 1, 2021 in India [13] and Songkran Buddhist new year celebration from April 12–15, 2021 in Thailand [14].

Individual-based simulation has traditionally been used to target problems in epidemics involving the mixing of heterogenous populations and time-varying contact rates. Assessing the influence of a mass gathering event on an epidemic falls into this type of problem, where infectious cases from outside may have a non-small effect on the $R_t$ even among local residents. In particular, the total number of visitors and the prevalence of infection among them, which may be comparable to the residents in the region, are key factors in such predictions. For this reason, we built an individual- (or agent)-based model with inflow and assessed how the growth of new cases could be controlled by different interventions under different prevalence rates among visitors.

## 2. Methods

### 2.1 Individual-based model

A mass gathering event yields additional gathering occasions, called secondary gatherings, that involve local and outside populations. For an appropriate description of transmission dynamics, the import of infected people between two types of temporarily formed populations should be considered. For this reason, we employed an individual-based description of the epidemic rather than a compartmental one such as the SIR model.

Specifically, we constructed an individual-based model by extending our study of the 2009 influenza pandemic [15] to deal with main and secondary gathering events. A simulation implementing the baseline model describes the activity of residents in a hypothetical city consisting of five towns that are tightly connected via commuter train, each of which being a collection of households, companies, schools, and shopping centers. Temporal population in these areas are formed by role-based activity of agents, classified into students, office workers, housekeepers, and visitors from outside the city. The state of an individual with respect to the disease is classified in the same manner as the SEIR model, that is, susceptible (S), infected but not yet infectious (E), infected and infectious (I), and recovered to gain a perfect immunity (R). In each local place, susceptible agents suffer infection risk $P$ during the small time period $\Delta t = 1\text{min}$ :

$$P = R_t \cdot \frac{\Delta t}{T_{\text{inf}}} \cdot \frac{I}{N},$$

where $N$ and $I$ are the (temporal) numbers of total and infectious visitors, respectively, $R_t$ is a parameter describing the transmissibility of this place, and $T_{\text{inf}} = 3 \text{ days}$ is the mean infectious period. The parameter $R_t$ is interpretable as the effective reproduction number in this place; in fact, if an infectious agent ( $I = 1$ ) and $N$ susceptible agents keep staying together for $T_{\text{inf}}$ , then the number of new infections for this duration is exponentially distributed and its mean is $R_t$ . An infected individual is expected to become infectious in a time that follows an exponential distribution with a mean of $T_{\text{latent}} = 3.5$ days. These settings are unified and formalized as transition probabilities between states

$$P(S \to E)\Delta t = R_t \cdot \frac{\Delta t}{T_{\text{inf}}} \cdot \frac{I}{N}, \; P(E \to I)\Delta t = \frac{\Delta t}{T_{\text{latent}}}, \; P(I \to R)\Delta t = \frac{\Delta t}{T_{\text{inf}}},$$

which reproduce the (stochastic version of) SEIR model if the model city consists of a single place.

The simulated city is designed as a simplified model of central Tokyo; a schematic illustration of this city is shown in Fig 1, and the numbers of entities are listed in Table 1. The number of populations and schools are chosen according to governmental statistics, while the other entities (shops, parks, and offices) are introduced uniformly to respective towns because of the difficulty in fitting to our simulator the real places with a greater diversity in scale. In total, about 1 million residents live in the city and have the chance to visit the main event, which is set to have 50,000 attendees. The timing of attendance is random.

The main mass gathering event consists of sub-venues, each of which randomly selected residents are scheduled to attend, and attendees in different sub-events are mutually segregated.

The simulated city is designed so that a set of places have only a threshold-level reproduction number, but an activity of visitors higher than residents could alter the tread in the number of new cases. Below, we briefly introduce the assumptions on the situation in the

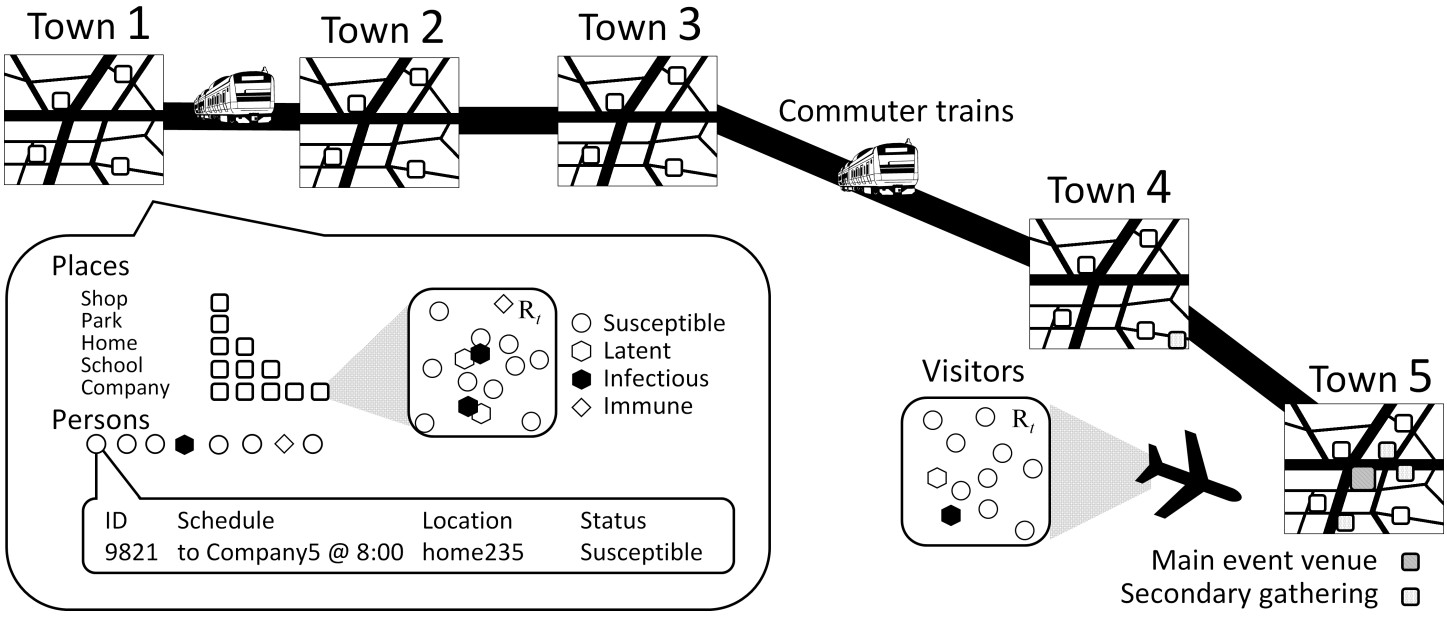

**Fig 1. Schematic illustration of the simulated city.**

**Table 1. Populations and number of places in the simulation.**

|  | Town 1 | Town 2 | Town 3 | Town 4 | Town 5 |
|---|---|---|---|---|---|
| Population | 571,641 | 176,866 | 138,684 | 314,861 | 44,680 |
| #Shops | 100 | 100 | 100 | 100 | 100 |
| #Parks | 2 | 2 | 2 | 2 | 2 |
| #Schools | 70 | 20 | 12 | 29 | 8 |
| #Offices | 100 | 100 | 100 | 2,000 | 2,000 |

simulated city and the reasoning behind them. It is unlikely that the epidemic in concern is highly reproducible when a mass gathering event is allowable. For this reason, the transmissibility parameters in populations in the simulated city have been configured so that the $R_t$ of the entire city is nearly equal to unity and the number of daily cases is kept almost constant until the event. The game venue is also not expected to greatly modify the trend in total number of new cases.

However, a non-small number of attendees would be expected to visit some other location after the event is over in order to meet with or socialize with one another, thereby creating many secondary gathering events. Some of these gatherings cause potentially the so-called super-spreading. Accordingly, game watchers in simulation are scheduled probabilistically to attend additional gathering with a variety infection risk there. Specifically, this calibration was carried out in a heuristic manner because unlike compartmental models, this critical condition is not described in a simple formula. We further assume that inbound attendees come to the main event multiple times and always join an additional event after the main one, considering that such travelers would necessarily eat out, in contrast to local attendees. The abovementioned assumptions are reflected in the parameter setting shown in Table 2, where the infection risk is quantified in terms of the $R_t$. In particular, a heavy tailed distribution of Rt at secondary gathering places is introduced to describe potential superspreading events. In

**Table 2. Parameters defining the activity of people in the simulation.**

| Reproduction number (R$t$) | | |
|---|---|---|
| Main events | 25 (baseline) or 1 (controlled) | |
| Secondary events (uncontrolled)* | 17-24 (90%), 240 (10%) | |
| Secondary events (controlled)* | | |
| Other sites | 1.05 | |
| | Domestic | Inbound |
| Prevalence | 0.01% | ≦1% (scenario-dependent) |
| Per capita attendance times | 1 | 1-3 |
| Probability of a seconcary event* | 0-0.1 (50%), 0.1-0.9 (40%), 0.9-1 (10%) | 1 |

* given as a piecewise uniform distribution: value range (frequency in %).

households, schools, and corporations, the risk is small, unlike the 2009 swine flu pandemic. Therefore, the $R_t$ values are set to a value close to unity. The attributed $R_t$ value, 1.05, to places other than those for the main and sub-events was chosen heuristically to establish a sustained (non-growing) epidemic. The proportion of the main event attendees who are dropped (e.g., to meet their friends) is provided by a probabilistic distribution.

## 2.2 Scenarios

To examine the possibility of suppressing the additional impetus to the epidemic brought by visitors, we carried out simulations with the following three scenarios:

1. **StadiumOnly:** The control is administered only in the main event venue. In the simulation, the control is implemented by a reduction in the $R_t$ from 25 to 1 (see Table 2).

2. **Extra:** In addition to StadiumOnly, secondary events are assumed to be controlled. This is implemented by reducing the proportion of highly risky places ($R_t$ = 240) for secondary events from 10% to 1%.

3. **WithoutDrop:** In addition to StadiumOnly, most of the attendees (specifically 90% in the simulation) are assumed to go directly home in accordance with a corresponding governmental campaign. The risk distribution in secondary events is the same as that in the StadiumOnly scenario.

The abovementioned values are arbitrary, but reflect the following assessment. First, in the simulations, we set the latent period to 3.5 days and the infectious period to 3 days, as average. We here assign rather shorter values, considering these periods are exponentially distributed in our model while the reality is close to a gamma or Weibull distribution [7]. Second, Murakami [3] assessed the infection risk in the Opening Ceremonies of the Tokyo 2020 Olympic Games as follows. An infectious attendee would yield about 1.5 or 0.01 new cases in uncontrolled and controlled settings, respectively, during a 5-hour-long ceremony, which is translated into the $\mathbf{R}_t$ by:

$$\mathbf{R}_t = (3 \text{days}) / (5/24 \text{days}) \times (1.5 \text{ or } 0.01) \text{ per capita cases} \approx 22 \text{ or } 0.1.$$

As for the controlled case, we used the threshold value, $R_t$ = 1, instead for this rather small risk. A similar assessment is available for another event, a 4-hour long soccer game: 0.74 or 0.022 cases are reproduced by one infectious case, respectively. A higher risk at some of the secondary gatherings is based on a survey targeting a large indoor convention [16]. That study

showed that respondents testing positive were more likely to have attended pubs, karaoke bars, or nightclubs. Using their reported values, attendees to these additional activities are at roughly three times higher risk of infection. Further assuming that this tripled risk is experienced within 1.5 hours, we take $R_t = 24 \times 5 / 1.5 \times 3 = 240$ to be the worst case scenario at additional meetings.

## 3. Results

Whether main mass gathering events accelerate epidemics depends on the baseline setting, the prevalence rate among visitors, and possible counter-measures to help prevent close contact. Fig 2 summarizes the outcome under each setting as an epidemic curve. As a baseline, the transmissibility parameters in respective small populations in the simulated city were configured so that the $R_t$ of the entire city remained nearly equal to unity until the event opened. As was configured, the number of new cases remained almost constant until the main event opened on day 14, and then tended to increase with a different slope until the event closed on day 24.

With the measure against only the event venue (scenario **StadiumOnly**) and zero infectious visitors, the new cases increased with a slope of $R_t$ = 1.08, as shown at the top left of Fig 2. This increasing trend was sustained when the prevalence of infection among the visitors was mildly high, $p_{\mathrm{inf,vis}} = 0.1\%$, but rapidly increased to $R_t$ = 1.33 when the prevalence of infection was extremely high, at $p_{\mathrm{inf,vis}} = 1\%$. These results suggest that cases imported from the outside would have only a minor impact if $p_{\mathrm{inf,vis}} < 0.1\%$.

Regarding secondary events under scenario **Extra**, it may be possible to erase the uptrend attributable to the main event when the visitors are fully noninfectious. Infectious visitors

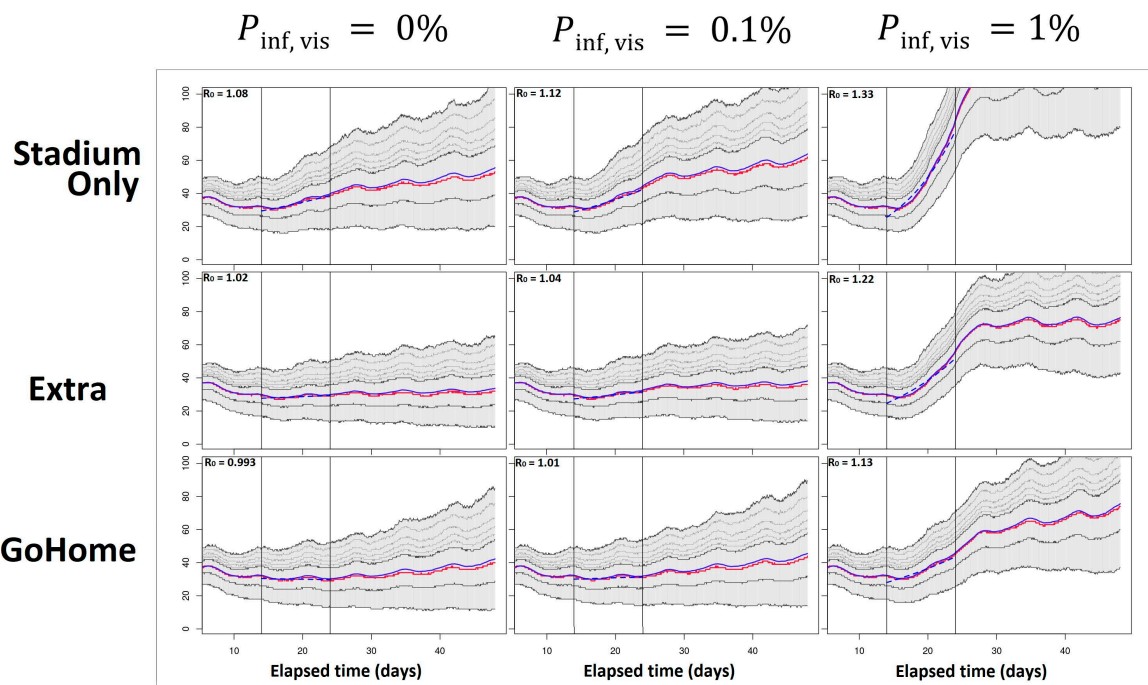

**Fig 2. Simulated epidemic curves for different assumptions on the prevalence rate among visitors and possible measures.** Thick lines indicate the mean (blue) and median (red) trajectories, while thin lines correspond to respective quantiles (25th, 75th, 80th, 85th, 90th, and 95th percentiles). The main mass gathering event runs from day 14 through 23 and is bounded by vertical lines.

may reinstate an uptrend, but our findings suggest that it would be rather weak as long as the prevalence was mild: $R_t = 1.04$ for $p_{\text{inf,vis}} = 0.1\%$. A recommendation to visitors to return directly home (scenario **WithoutDrop**) would have a similar effect. A constant trend would be sustained after the start of the event start if no infectious visitors came in, or a shift to a weak uptrend if their prevalence rate was $< 0.1\%$.

So far, we have interpreted the impacts of infectious visitors and controls, viewing only the mean epidemic curve trajectories. However, these curves are highly stochastic, as shown in the gray shading in each panel of Fig 2, and the slope of the trend involves a greater uncertainty, as shown in Fig 3. Fig 4 shows the $R_t$ with an uncertainty for all scenarios as a function of $p_{\text{inf,vis}}$. The strongest measure of **WithoutDrop** suppresses the increase in the $R_t$ based on the prevalence rate of visitors. On average, $p_{\text{inf,vis}} \leq 0.1\%$ keeps the $R_t$ almost at unity under this measure.

## 4. Discussion and Conclusion

In this study, we demonstrated how the potential growth of an epidemic can be successfully or unsuccessfully suppressed in a mass gathering event involving a comparable number of local and outside visitors depending on the strength of the intervention and the prevalence rate among visitors. As a general remark, the growth rate potentially enhanced by hosting a mass gathering can be suppressed to a level identical or slightly higher than that of the baseline if the places for gathering after the main event are well controlled and the prevalence rate among visitors is mild relative to that among local residents.

The controls discussed in this work are voluntary-basis and hence the degree people are willing to cooperate limits the effectiveness. A massive superspreading along with

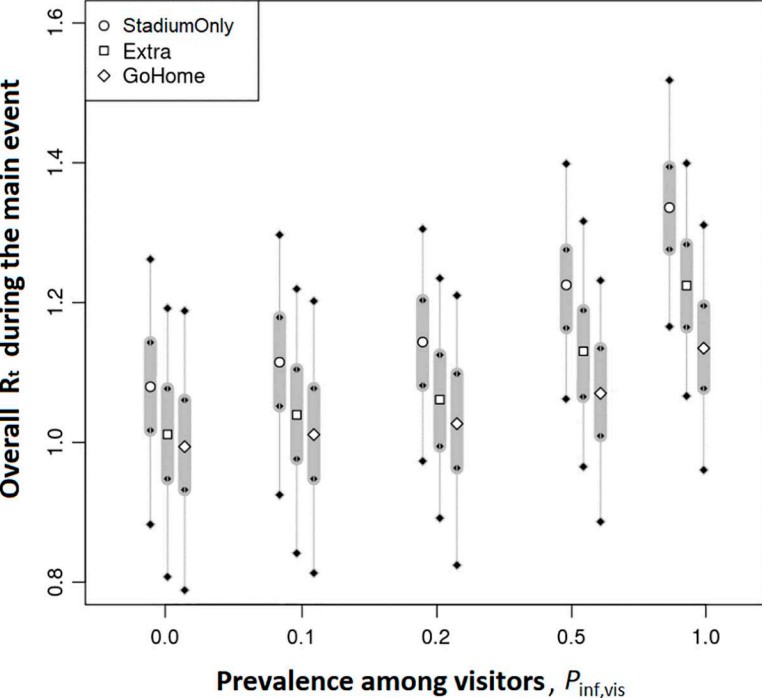

**Fig 3. The effective reproduction number ($Rt$) in the entire city during the main event in each scenario.** The uncertainty in each setting is evaluated from 1,000 runs with a different random number sequence.

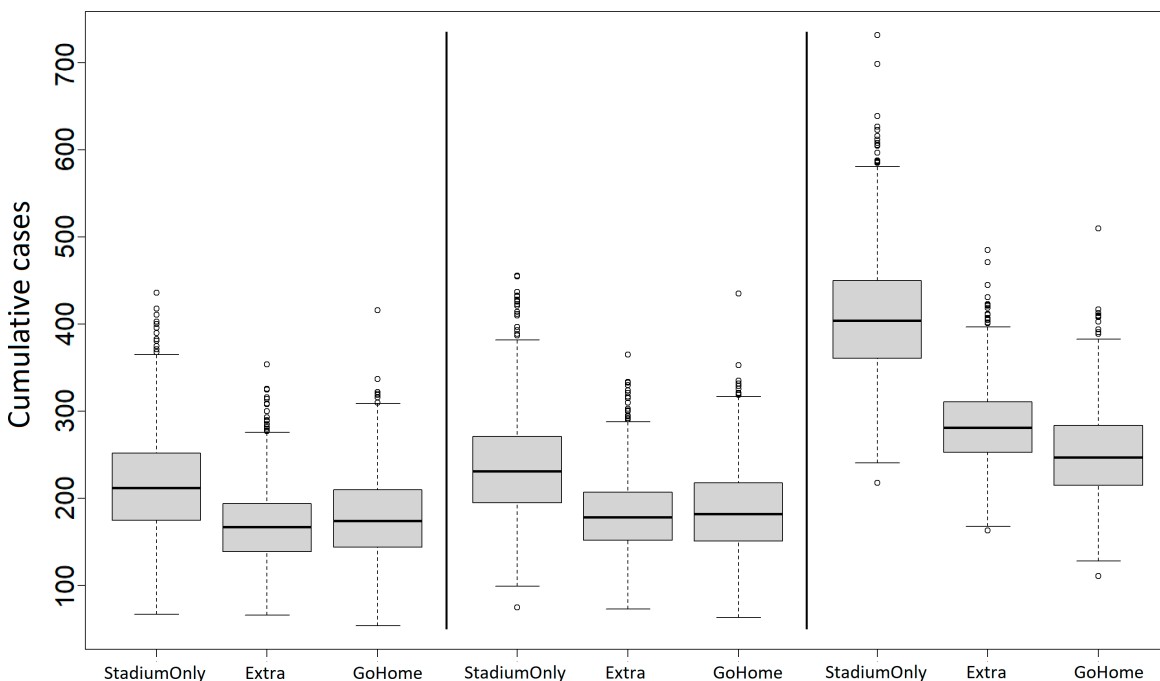

**Fig 4. Distributions of cumulative cases during the event corresponding to** Fig 2**. The uncertainty is based on 1,000 simulation runs.**

Kumbh Mela [13] religious could be numbered as a case that such a cooperation is poorly established, and many attendees reportedly did not follow to the requests of mask wearing and social distancing. Though we are not sure how to measure directly such a willingness via investigation, we can refer some articles which help future discussion. People yielding higher score in willingness to cooperate may not show a greater acceptance of behavioral regulations [17]. The cultural difference affects on which of individual-basis or law-enforced measures are preferable. A comparative study of three countries [18] shows Chinese participants are more acceptable to enforced measures than Japan and US, while US and Chinese more accept individual-level measures than does Japanese. Another investigation [19] supports that Japanese people are reluctant to enforced restrictions in a greater degree than the other participating countries. As for mask wearing, one of self-motivated measures, there are studies over 50 US states and over the world that support people in collectivistic communities more frequently wear masks than in individual ones [20]. Also, it is possible to build a cooperative relationship among supports fan, academic experts, and event organizers: as high as about 98% of chant cheering participants wore masks in games held by Japan Football leagues [21].

Reflecting the lack of qualitatively interpretable data for shops, restaurants, and other kinds of possible meeting places, we carried out simulations with scenario-based settings to assess risk in these places. Below, we discuss the availability of data from the parameters of our simulation. The Japanese government enacted a strict quarantine at airports during the pandemic. The data collected from the quarantine [22,23] could be expected to aid the estimation of prevalence rates among visitors, thereby making quarantine more effective in combination with testing before boarding and after arrival [24].

In addition to the prevalence obtained along with the quarantine, the prevalence rate among the participants of the Tokyo 2020 Olympic Games is available [25]. Maintaining a sufficient level of control at secondary gathering places is more difficult than the main event venue. The Tokyo Metropolitan Government issued an "Infection Prevention Thorough Declaration Sticker" to shops, restaurants, and other meeting places to establish a certain level of risk reduction and provide information on well-sanitized places to local residents [26]. Hence, the ratio of safe places to 'risky places' (10% or 1% being assumed in the present study) may be determined via data. However, the requirements for this are mainly qualitative, and it is nontrivial to translate these data into the risk of infection computationally interpretable.

Observational facts that contribute to connecting the emission of droplets to the risk of infection have accumulated since the beginning of the COVID-19 pandemic and the abovementioned difficulties may be overcome by plugging these data into simulations. While some of this knowledge has yet to be fully authorized, a number of studies have already been published. Ando et al. [27] assessed infection risk at a restaurant using a dedicated hydrodynamical model on the Fugaku supercomputer. The effects of air ventilation in the venue have been monitored based on $CO_2$ concentrations during a football game [28], and droplet emissions have been measured during normal speech [29], coughing [30], and singing [31]. In addition, a COVID-19 outbreak event suspected to be induced by air conditioning in a restaurant has been reported [32]. We expect that a reliable set of infection risk parameters will become available based on the outcomes of empirical studies interpreted inside hydrodynamical droplet simulations, and that individual-based models will contribute to improved risk assessments against new variants of severe acute respiratory syndrome coronavirus 2 and future pandemics caused by other infectious diseases.

## Author contributions

**Conceptualization:** Masaya M. Saitoh, Kotoe Katayama, Michio Murakami, Tomoaki Okuda, Tetsuo Ysutaka, Masaharu Tsubokura, Seiya Imoto.

**Formal analysis:** Akira Naruse, Peiying Ruan, Seiya Imoto.

**Investigation:** Masaya M. Saitoh.

**Methodology:** Masaya M. Saitoh, Michio Murakami.

**Project administration:** Kotoe Katayama, Seiya Imoto.

**Software:** Akira Naruse, Peiying Ruan.

**Supervision:** Michio Murakami.

**Visualization:** Masaya M. Saitoh.

**Writing – original draft:** Masaya M. Saitoh.

**Writing – review & editing:** Kotoe Katayama, Akira Naruse, Peiying Ruan, Michio Murakami, Tomoaki Okuda, Tetsuo Ysutaka, Wataru Naito, Masaharu Tsubokura, Seiya Imoto.

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
