## [Decision Letter · Decision Letter 0]

10 Jul 2023

PONE-D-23-04678Effects of inbound attendees of a mass gathering event on the COVID-19 epidemic using individual-based simulationsPLOS ONE

Dear Dr. Saitoh,

Thank you for submitting your manuscript to PLOS ONE. After careful consideration, we feel that it has merit but does not fully meet PLOS ONE’s publication criteria as it currently stands. Therefore, we invite you to submit a revised version of the manuscript that addresses the points raised during the review process.

ACADEMIC EDITOR: Kindly address the comments and suggestions of the 3 reviewers. Please ensure that your decision is justified on PLOS ONE’s publication criteria  and not, for example, on novelty or perceived impact.

We look forward to receiving your revised manuscript.

Kind regards,

Ian Christopher N Rocha, MD, MBA, MHSS

Academic Editor

PLOS ONE

Journal Requirements:

"This work was supported by JSPS KAKENHI Grant Numbers JP18K11541."

"One of the authors, Masaya M. Saito was supported by JSPS KAKENHI Grant Numbers JP18K11541."

"One of the authors, Masaya M. Saito was supported by JSPS KAKENHI Grant Numbers JP18K11541."

4. We noted in your submission details that a portion of your manuscript may have been presented or published elsewhere. [The manuscript was registered in the preprint server medRXiv (https://www.medrxiv.org/content/10.1101/2022.08.31.22279428v1). As a cumstom, the preprint server's role is considered to just provide the access to the documents to researchers , not to authorizing them as publications of the organization running the server. For this reason, we consider that the manuscript we are submitting has not been published in advance.] Please clarify whether this [conference proceeding or publication] was peer-reviewed and formally published. If this work was previously peer-reviewed and published, in the cover letter please provide the reason that this work does not constitute dual publication and should be included in the current manuscript.

Upon re-submitting your revised manuscript, please upload your study’s minimal underlying data set as either Supporting Information files or to a stable, public repository and include the relevant URLs, DOIs, or accession numbers within your revised cover letter. For a list of acceptable repositories, please see http://journals.plos.org/plosone/s/data-availability#loc-recommended-repositories . Any potentially identifying patient information must be fully anonymized.

Important: If there are ethical or legal restrictions to sharing your data publicly, please explain these restrictions in detail. Please see our guidelines for more information on what we consider unacceptable restrictions to publicly sharing data: http://journals.plos.org/plosone/s/data-availability#loc-unacceptable-data-access-restrictions . Note that it is not acceptable for the authors to be the sole named individuals responsible for ensuring data access.

Additional Editor Comments :

Kindly address the comments of the 3 reviewers. Thank you.

Reviewers' comments:

Reviewer's Responses to Questions

**Comments to the Author**

1. Is the manuscript technically sound, and do the data support the conclusions?

Reviewer #1: Yes

Reviewer #2: Partly

Reviewer #3: Yes

2. Has the statistical analysis been performed appropriately and rigorously? 

Reviewer #1: Yes

Reviewer #2: Yes

Reviewer #3: Yes

3. Have the authors made all data underlying the findings in their manuscript fully available?

Reviewer #1: Yes

Reviewer #2: Yes

Reviewer #3: Yes

4. Is the manuscript presented in an intelligible fashion and written in standard English?

Reviewer #1: Yes

Reviewer #2: Yes

Reviewer #3: Yes

5. Review Comments to the Author

Reviewer #1: Thank you for submitting your article on the potential measures to prevent the acceleration of an outbreak of an infectious disease during mass gathering events. While your research presents important findings, I would like to request some clarification on certain points to help strengthen your work.

Could you please provide further detail on how the individual-based simulator was built and the specific assumptions used in the model?

Can you elaborate on the reasons behind the choice of the hypothetical city used in the simulation?

Were any limitations or assumptions made regarding the potential spread of the infectious disease beyond the event and the city?

While the results of your study suggest that the increase in the number of COVID-19 patients can be suppressed to a level equivalent to that if the event were not being held, can you discuss any potential challenges or limitations to implementing the suggested controls and restrictions on visitors?

Your responses to these questions will help provide a deeper understanding of your research and contribute to the broader scientific discourse.

Thank you for your attention to these points.

Reviewer #2: Review Report

Title: Effects of inbound attendees of a mass gathering event on the COVID-19 epidemic using individual-based simulations

Manuscript ID: PONE-D-23-04678.

Review Comments

The study methos for assessing and predicting the transmission of the COVID-19, the settings and contexts and the whole means seems inappropriate with the way researches assess and predict the transmission of new respiratory pandemics E.g., mobility.

Regards,

Reviewer #3: Dear Authors,

Thank you for submitting an interesting manuscript. After thorough review, I am recommending some revisions. In this regard, kindly address the following comments and suggestions to further improve your manuscript:

1) In this paper, the authors identified that the effects of inbound attendees of a mass gathering event. The authors were also able to present and demonstrate their findings very well. I congratulate them for doing a good job.

2) A mass gathering event can be superspreading events (SSE) and it has a fairly formal definition in infectious disease epidemiology literature, first requiring an estimate of R, and then identifying events where the number of secondary infections from a single case is substantially above the average (i.e. substantially above R), a minimum threshold might be at least 2x R or 3x R. In this regard, I would like you to read on this paper and cite to further improve the introduction and discussion sections, https://doi.org/10.1080/03007995.2022.2125258

3) Additionally, the authors may also use these papers in describing and citing SSEs from other countries, such as what happened in India (the Kumbh Mela massive SSE which infected millions of people) https://doi.org/10.4269/ajtmh.21-0601 and in Thailand (the scattered SSEs during holidays) https://doi.org/10.34172/ijtmgh.2021.33

4) It is also nice to include some recommendations and to further improve the conclusion.

Thank you and best regards,

6. PLOS authors have the option to publish the peer review history of their article (what does this mean? ). If published, this will include your full peer review and any attached files.

**Do you want your identity to be public for this peer review?** For information about this choice, including consent withdrawal, please see our Privacy Policy .

Reviewer #1: No

Reviewer #2: No

Reviewer #3: No

---

## [Author Response · Author response to Decision Letter 1]

6 Feb 2024

Reply to Reviewer 1

Could you please provide further detail on how the individual based simulator was built and the specific assumptions used in the model?

Can you elaborate on the reasons behind the choice of the hypothetical city used in the simulation?

[answer] Our main interest is to assess whether additional gathering accompanying the main event enhance the reproduction. Emphasizing this point, we have rephrased the paragraphs in section 2.1 Individual-based model.

Were any limitation or assumptions made regarding the potential spread of the infectious disease beyond the event and the city?

[answer] In this study, we haven’t discussed in detail how degree people are willing to corporate the recommendation. Along with your next remark, we have add a discussion section for this topic.

While the results of your study suggest that the increase in the number of COVID-19 patients can be suppressed to a level equivalent to that if the event were not being held, can you discuss any potential challenges or limitations to implementing the suggested controls and restrictions on visitors?

[answer] The controls discussed in this work are voluntary-basis and their effectiveness depends on how degree the participants are willing to cooperate. We refer several articles which do not directly evaluate this but would help us to discuss this problem and add the sentences below:

The controls discussed in this work are voluntary-basis and hence the degree people are willing to cooperate limits the effectiveness. People yielding higher score in willingness to cooperate may not show a greater acceptance of behavioral regulations [17].

The cultural difference affects on which of individual-basis or law-enforced measures are preferable. A comparative study of three countries [18] shows Chinese participants are more acceptable to enforced measures than Japan and US, while US and Chinese more accept individual-level measures than does Japanese. Another investigation [19] supports that Japanese people are reluctant to enforced restrictions in a greater degree than the other participating countries.

As for mask wearing, one of self-motivated measures, there are studies over 50 US states and over the world that support people in collectivistic communities more frequently wear masks than in individual ones [20].

Also, it is possible to build a cooperative relationship among supports fan, academic experts, and event organizers: as high as about 98% of chant cheering participants wore masks in games held by Japan Football leagues [21].

Replay to Reviewer 3

2) A mass gathering event can be superspreading events (SSE) and it has a fairly formal definition in infectious disease epidemiology literature, first requiring an estimate of R, and then identifying events where the number of secondary infections from a single case is substantially above the average (i.e. substantially above R), a minimum threshold might be at least 2x R or 3x R. In this regard, I would like you to read on this paper and cite to further improve the introduction and discussion sections,

https://doi.org/10.1080/03007995.2022.2125258

3) Additionally, the authors may also use these papers in describing and citing SSEs from other countries, such as what happened in India (the Kumbh Mela massive SSE which infected millions of people) https://doi.org/10.4269/ajtmh.21-0601 and in Thailand (the scattered SSEs during holidays) https://doi.org/10.34172/ijtmgh.2021.33

[answer] Thank you for giving a remark on the importance of superspreading and related references. To address your suggestion, we have added a topic addressing your suggestion on superspreading in Introduction.

4) It is also nice to include some recommendations and to further improve the conclusion.

[answer] I have added a paragraph in Conclusion section to discuss how degree voluntary based measures are accepted. The Kumbh Mela massive SSE case is again referred here as a case study where a cooperation among participants is not well established.

Changes in reference list

I below show the changes in the reference list. The number to each reference is of the present version (i.e. renumbered from the original submission).

Added articles in this revision: 7, 8, 9, 10, 11, 12, 17, 18, 19, 20, and 21.

Removed articles in this revision: none.

Articles whose reference is modified: 26

Reason: The original URL in a governmental web site is no longer available and replaced to wayback machine’s URL.

---

## [Decision Letter · Decision Letter 1]

1 Dec 2024

PONE-D-23-04678R1Effects of inbound attendees of a mass gathering event on the COVID-19 epidemic using individual-based simulationsPLOS ONE

Dear Dr. Saitoh,

Thank you for submitting your manuscript to PLOS ONE. After careful consideration, we feel that it has merit but does not fully meet PLOS ONE’s publication criteria as it currently stands. Therefore, we invite you to submit a revised version of the manuscript that addresses the points raised during the review process.

Because of the conflicting opinions of Reviewers 1 and 2, we have engaged an additional reviewer. Please address the comments by Reviewer 4, and provide a point-by-point rebuttal.

We look forward to receiving your revised manuscript.

Kind regards,

Siew Ann Cheong, Ph.D.

Academic Editor

PLOS ONE

Journal Requirements:

Reviewers' comments:

Reviewer's Responses to Questions

**Comments to the Author**

1. If the authors have adequately addressed your comments raised in a previous round of review and you feel that this manuscript is now acceptable for publication, you may indicate that here to bypass the “Comments to the Author” section, enter your conflict of interest statement in the “Confidential to Editor” section, and submit your "Accept" recommendation.

Reviewer #1: All comments have been addressed

Reviewer #2: All comments have been addressed

Reviewer #4: (No Response)

2. Is the manuscript technically sound, and do the data support the conclusions?

Reviewer #1: Yes

Reviewer #2: Partly

Reviewer #4: Partly

3. Has the statistical analysis been performed appropriately and rigorously? 

Reviewer #1: Yes

Reviewer #2: Yes

Reviewer #4: No

4. Have the authors made all data underlying the findings in their manuscript fully available?

Reviewer #1: Yes

Reviewer #2: Yes

Reviewer #4: No

5. Is the manuscript presented in an intelligible fashion and written in standard English?

Reviewer #1: Yes

Reviewer #2: Yes

Reviewer #4: Yes

6. Review Comments to the Author

Reviewer #1: (No Response)

Reviewer #2: Review Reports

Title :Effects of inbound attendees of a mass gathering event on the COVID-19 epidemic using

individual-based simulations

Review Comments

-The title should be in line with the scope of the study E.g. Is thar transmission , infectiousness or morbidity or what type of study?

-If WHO have scenarios approach to project the rate of transmission and possible morbidity, and If SEIRS approach is in place what new body of knowledge can a reader grasp?

-The background is slightly I can guess not actually a background of scientific paper.

- On the methods section:

-The scenario's are incomplete e.g. the second scenrio and the magnitude and probality of visiting other area E.g. Shops.

- Tge background assumption is lacking E.g. The magnitude of the pandemic and the wave at which Japan is facing at that time

-Contact and tge way they contact each other as well as the use of PPEs is missed/The whether use of PPEs

-HICs Vs infectious disease prevalence is also missed.

- Variation by other sociodemographic variabke should be also imputed e.g. indiginious Vs visitors. Probability of use of train and the ctual distance travelled in the hyoothetical city should be empahsized.

-Lacks practical considerations E.g. estimated number of population in the five hypothetical towns..etc

- The resut is too short and lacks full explanation. E.g You are expected to put at least one figure in the main documents .

-The dicussion is short and mainly focused on the limitations and recommendations of qulaitative approach .

-In general it is incomplete E.g. the author's contributions, conflict of interest is is lacking

Regards,

Reviewer #4: Dear author

Some issues have been identified in your manuscript. Please refer to the annotated details in the enclosed document.

Thank You

7. PLOS authors have the option to publish the peer review history of their article (what does this mean? ). If published, this will include your full peer review and any attached files.

**Do you want your identity to be public for this peer review?** For information about this choice, including consent withdrawal, please see our Privacy Policy .

Reviewer #1: **Yes: ** Nguyen Tran Minh Duc

Reviewer #2: No

Reviewer #4: **Yes: ** Norhayati Rosli

---

## [Author Response · Author response to Decision Letter 2]

2 Mar 2025

The requested rebuttal to comments raised by Reviewer 4 is included in the file 'Response_to_Reviewers_table.xlsx'.

---

## [Editor Report · Decision Letter 2]

4 Mar 2025

Effects of inbound attendees of a mass gathering event on the COVID-19 epidemic using individual-based simulations

PONE-D-23-04678R2

Dear Dr. Saitoh,

We’re pleased to inform you that your manuscript has been judged scientifically suitable for publication and will be formally accepted for publication once it meets all outstanding technical requirements.

Kind regards,

Siew Ann Cheong, Ph.D.

Academic Editor

PLOS ONE
---

## [Editor Report · Acceptance letter]

PONE-D-23-04678R2

PLOS ONE

Dear Dr. Saitoh,

I'm pleased to inform you that your manuscript has been deemed suitable for publication in PLOS ONE. Congratulations! Your manuscript is now being handed over to our production team.

Kind regards,

on behalf of

Dr. Siew Ann Cheong

Academic Editor

PLOS ONE